# Predicting soft tissue thicknesses overlying the iliac crests and greater trochanters of younger and older adults

Claudia M. Town *©, Danielle L. Gyemi©, Zoe Ellis‡, Charles Kahelin‡, Andrew C. Laing©, David M. Andrews©

Department of Kinesiology, University of Windsor, Windsor, Ontario, Canada

© These authors contributed equally to this work.
‡ ZE and CK also contributed equally to this work.
* town1@uwindsor.ca

**Data Availability Statement:** All relevant data are within the paper and its Supporting Information files.

## Abstract

Soft tissues overlying the hip play a critical role in protecting against fractures during fall-related hip impacts. Consequently, the development of an efficient and cost-effective method for estimating hip soft tissue thicknesses in living people may prove to be valuable for assessing an individual's injury risk and need to adopt preventative measures. The present study used multiple linear stepwise regression to generate prediction equations from participant characteristics (i.e., height, sex) and anthropometric measurements of the pelvis, trunk, and thigh to estimate soft tissue thickness at the iliac crests (IC) and greater trochanters (GT) in younger (16–35 years of age: 37 males, 37 females) and older (36–65 years of age: 38 males, 38 females) adults. Equations were validated against soft tissue thicknesses measured from full body Dual-energy X-ray Absorptiometry scans of independent samples (younger: 13 males, 13 females; older: 13 males, 12 females). Younger adult prediction equations exhibited adjusted $R^2$ values ranging from 0.704 to 0.791, with more explained variance for soft tissue thicknesses at the GT than the IC; corresponding values for the older adult equations were higher overall and ranged from 0.819 to 0.852. Predicted and actual soft tissue thicknesses were significantly correlated for both the younger ($R^2$ = 0.466 to 0.738) and older ($R^2$ = 0.842 to 0.848) adults, averaging ≤ 0.75cm of error. This research demonstrates that soft tissue thicknesses overlying the GT and IC can be accurately predicted from equations using anthropometric measurements. These equations can be used by clinicians to identify individuals at higher risk of hip fractures who may benefit from the use of preventative measures.

## Introduction

Over 90% of hip fractures are the result of a fall [1], most of which occur in a sideways direction directly to the hip or side of the leg [2, 3]. The risk of fall-related hip fractures is especially high among older adults and is cited as a primary cause of morbidity and mortality in this

**Funding:** DA NSERC Discovery Grant Number RGPIN/227682 Natural Sciences and Engineering Research Council of Canada https://www.nserc-crsng.gc.ca/index_eng.asp No, the funders did not play any role in the study design, data collection and analysis, decision to publish, or preparation of the manuscript.

**Competing interests:** The authors have declared that no competing interests exist.

population [4, 5]. While less severe hip injuries (hip pointers and contusions) are more common in younger adults due to sport-related hip impacts [6], they may also experience hip fractures from a lateral fall [7]. Given that hip fractures can cost healthcare systems billions of dollars each year [8], research efforts have focused on developing protective strategies and devices to help mitigate the risk of experiencing injurious hip impacts [9–12].

The soft tissue thickness overlying specific bony regions of the hip, such as the greater trochanter (GT), has been reported to provide natural protection against fall-related hip fracture by attenuating injurious impact forces [13, 14]. Therefore, quantifying soft tissue thicknesses in the hip region could help identify individuals who may be at greater risk of hip injury and fracture [13–16]. Trochanteric soft tissue thickness in older adults has previously been quantified from whole body Dual-energy X-ray Absorptiometry (DXA) scans by manually measuring the distance between the GT and lateral aspect of the skin-air boundary [13, 17]. These measurements were positively correlated (r = 0.75) with body mass index (BMI) in both women [13] and men [17]. Dufour et al. (2012) later applied these sex-specific regression equations to estimate trochanteric soft tissue thicknesses when assessing the validity of the factor-of-risk method [18] in hip fracture prediction [19]. Schacter and Leslie (2014) used multiple linear regression to develop a trochanteric soft tissue thickness prediction equation from regional anteroposterior DXA scans performed for osteoporosis risk assessment, wherein direct measurement is not possible given the scan's limited focus on the lumbar spine and hip regions. Variables used to generate the prediction equation included sex, BMI, and two DXA-derived internal tissue thickness measures (i.e., spine average thickness and hip average thickness). This resulted in an adjusted $R^2$ value of 0.60, which was reported to have significantly better predictive capabilities compared to only using sex and BMI without any DXA measures [20].

With the exception of the BMI-driven equation presented by Dufour et al. (2012), current methods of directly measuring or indirectly estimating hip soft tissue thicknesses are heavily reliant on the use of DXA [13, 17, 20]. While DXA is a valuable tool that can provide accurate estimates of soft and bone tissue masses, it is a costly piece of equipment which is not easily accessible for many outside of clinical environments, and while fairly low in magnitude, DXA scanning does expose patients to radiation [20]. Being able to accurately predict hip soft tissue thicknesses for people across a range of ages without the use of DXA, would be useful for clinicians and would provide researchers with an accessible and less expensive option for estimating these values. Prior work has shown that soft and rigid tissue masses of body segments (upper and lower limbs, head, neck, trunk, and pelvis) in both younger and older adults can be predicted very accurately using regression equations based on easily measured anthropometric measurements [21–24]. Therefore, the purpose of the present study was to expand upon previous research by developing and validating a set of regression equations that utilize surface anthropometric measurements to predict bilateral soft tissue thicknesses overlying the iliac crests (IC) and GT of younger and older adults.

## Methods

### Participants

Data were collected on two separate participant groups: 100 healthy younger adults (50 female, 50 male; age 16–35 years) with a mean [SD] age, mass, and height of 24.5 [4.0] years, 70.6 [13.5] kg, and 1.71 [0.09] cm, and 101 healthy older adults (51 male, 50 female; 36–65 years) with a mean [SD] age, mass, and height of 49.2 [7.7] years, 78.1 [17.1] kg, and 1.70 [0.10] cm, respectively (Tables 1 and 2). Participants were excluded from this study if they were taller than 1.98 m or were pregnant. The height restriction was due to the length of the DXA scanning bed used in the study. Participants provided informed written consent prior to being

**Table 1. Mean (± SD) general physical characteristics and anthropometric measures for young (16–35 years) male and female participants in both generation and validation samples.**

| Variable/measure | Generation sample (*n* = 74) | | | | Validation sample (*n* = 26) | | | |
| --- | --- | --- | --- | --- | --- | --- | --- | --- |
| | Males (*n* = 37) | | Females (*n* = 37) | | Males (*n* = 13) | | Females (*n* = 13) | |
| Physical characteristics | | | | | | | | |
| Age (yrs) | 24.3 | (3.8) | 25.5 | (4.1) | 23.7 | (3.4) | 23.3 | (2.9) |
| Height (m) | 1.78* | (0.08) | 1.65 | (0.07) | 1.77 | (0.07) | 1.65 | (0.07) |
| Mass (kg) | 79.1* | (11.4) | 61.3 | (9.1) | 82.3 | (11.8) | 61.3 | (9.6) |
| Lengths (cm) | | | | | | | | |
| Pelvis (A) | 15.7 | (3.0) | 15.8 | (2.7) | 15.2 | (1.8) | 17.4 | (2.9) |
| Pelvis (L)** | 13.1* | (1.5) | 11.9 | (3.8) | 13.2 | (1.4) | 11.9 | (3.9) |
| Thigh (L)** | 55.3* | (3.2) | 51.0 | (5.4) | 53.7 | (4.9) | 50.5 | (5.5) |
| Thigh (M)** | 46.1* | (3.1) | 42.1 | (3.6) | 46.4 | (3.3) | 41.9 | (3.8) |
| Mid-Thigh** | 16.3 | (3.7) | 17.5 | (3.8) | 17.0 | (3.1) | 17.7 | (4.6) |
| Circumferences (cm) | | | | | | | | |
| Hips | 98.9* | (4.9) | 94.9 | (7.3) | 100.1 | (4.5) | 94.9 | (8.1) |
| Pelvis | 87.4* | (7.0) | 82.6 | (6.6) | 88.4 | (7.4) | 82.4 | (6.7) |
| Waist | 85.5* | (6.5) | 74.5 | (5.8) | 87.7 | (7.6) | 73.7 | (5.4) |
| Upper Thigh** | 60.7 | (4.5) | 60.1 | (4.2) | 62.3 | (3.8) | 59.5 | (4.3) |
| Mid-Thigh** | 57.6 | (4.6) | 55.7 | (4.0) | 59.0 | (3.4) | 55.1 | (2.8) |
| Breadths (cm) | | | | | | | | |
| Pelvis (A-P) | 18.5 | (1.6) | 18.3 | (2.1) | 18.8 | (0.8) | 18.5 | (2.4) |
| Waist (A-P) | 20.3* | (2.0) | 18.1 | (1.9) | 21.2 | (1.8) | 17.8 | (1.2) |
| Waist (M-L) | 30.9* | (2.3) | 26.4 | (1.9) | 31.4 | (2.5) | 26.4 | (1.8) |
| Hip (M-L) | 34.3 | (1.7) | 33.6 | (2.5) | 34.6 | (1.4) | 33.6 | (2.9) |
| Upper Thigh (A-P)** | 18.1 | (1.8) | 18.0 | (1.5) | 19.2 | (1.7) | 18.1 | (1.5) |
| Max Thigh (M-L)** | 17.0 | (1.5) | 16.9 | (1.4) | 17.6 | (1.1) | 17.0 | (1.2) |
| Max Thigh (A-P)** | 17.6* | (1.6) | 16.7 | (1.3) | 18.2 | (1.4) | 16.8 | (1.2) |
| Skinfolds (mm) | | | | | | | | |
| Suprailiac *** | 14.9 | (8.0) | 13.0 | (5.7) | 18.7 | (8.4) | 13.9 | (5.4) |
| Mid-Thigh (A)** | 12.9* | (8.2) | 25.6 | (11.1) | 14.2 | (6.8) | 25.1 | (6.6) |
| Mid-Thigh (P) ** | 12.7* | (6.0) | 27.3 | (10.5) | 12.9 | (4.7) | 26.5 | (8.7) |
| Abdomen*** | 18.9 | (10.9) | 19.4 | (9.0) | 24.0 | (10.0) | 21.1 | (8.0) |

*Note*. A = anterior; P = posterior; M = medial; L = lateral.

*$P < .05$, significant difference between sexes within generation sample.

**Average of lateral measurements from the left and right sides of the body.

*** Measurements were taken only from the right side of the body.

scanned and measured. This study was approved by the Research Ethics Boards of the affiliated university and hospital.

## Study procedures and instrumentation

Thirty-two anthropometric measurements (Tables 1 and 2: 9 lengths, 7 circumferences, 10 breadths, 6 skinfolds) from the trunk, pelvis and thigh segments of each participant were taken by two investigators while participants stood in the anatomical position. Measurements (described in Table 3) were acquired using standard tools: a flexible measuring tape, anthropometers (Layfayette Instrument Company, Layfayette, IN) and skinfold callipers (Slimguide®,

**Table 2. Mean (± SD) general physical characteristics and anthropometric measures for older (36–65 years) male and female participants in both generation and validation samples.**

| Variable/measure | Generation sample (*n* = 76) | | | | Validation sample (*n* = 25) | | | |
| --- | --- | --- | --- | --- | --- | --- | --- | --- |
| | Males (*n* = 38) | | Females (*n* = 38) | | Males (*n* = 13) | | Females (*n* = 12) | |
| Physical characteristics | | | | | | | | |
| Age (yrs) | 49.4 | (8.5) | 49.3 | (6.7) | 50.6 | (8.8) | 47.8 | (7.9) |
| Height (m) | 1.78* | (0.07) | 1.62 | (0.07) | 1.77 | (0.06) | 1.64 | (0.04) |
| Mass (kg) | 85.8* | (17.0) | 69.0 | (12.6) | 85.4 | (12.2) | 70.8 | (17.9) |
| Lengths (cm) | | | | | | | | |
| Pelvis (A) | 14.8 | (2.9) | 15.5 | (3.5) | 15.5 | (3.0) | 17.5 | (3.1) |
| Pelvis (L)** | 12.6* | (1.8) | 10.5 | (2.7) | 12.4 | (2.1) | 11.0 | (2.6) |
| Thigh (L)** | 54.3* | (4.1) | 49.2 | (4.7) | 55.5 | (3.1) | 51 | (4.4) |
| Thigh (M)** | 46.6* | (3.7) | 41.8 | (2.7) | 46.4 | (3.5) | 41.9 | (3.8) |
| Mid-Thigh** | 17.4 | (3.5) | 17.3 | (3.4) | 17 | (2.7) | 15.6 | (2.2) |
| Circumferences (cm) | | | | | | | | |
| Hips | 101.4 | (8.1) | 102.9 | (9.3) | 102.1 | (6.2) | 101.1 | (14.0) |
| Pelvis | 95.3 | (11.7) | 95.9 | (10.8) | 95.6 | (11.1) | 93.9 | (18.6) |
| Waist | 95.4* | (12.9) | 87.9 | (13.5) | 96.3 | (12.0) | 85.7 | (17.4) |
| Upper Thigh** | 61.4 | (5.4) | 62.3 | (5.7) | 60.5 | (5.6) | 62.2 | (7.9) |
| Mid-Thigh** | 56.4 | (4.9) | 56.7 | (5.5) | 56.5 | (5.1) | 57.4 | (7.6) |
| Breadths (cm) | | | | | | | | |
| Pelvis (A-P) | 19.8 | (2.5) | 19.8 | (2.5) | 19.5 | (2.6) | 18.7 | (3.2) |
| Waist (A-P) | 23.9 | (4.6) | 22.8 | (4.2) | 24.8 | (4.3) | 22.3 | (6.0) |
| Waist (M-L) | 33.4* | (3.5) | 30.4 | (3.5) | 33.8 | (3.2) | 29.7 | (5.2) |
| Hip (M-L) | 35.7 | 2.3 | 36.1 | 3.3 | 36.1 | (2.4) | 35.4 | (3.9) |
| Upper-Thigh (A-P)** | 18.4 | (2.1) | 18.3 | (2.0) | 18.2 | (2.8) | 18.7 | (2.6) |
| Max-Thigh (M-L)** | 16.7 | (1.7) | 17.5 | (1.9) | 17.1 | (1.8) | 17.7 | (2.1) |
| Max-Thigh (A-P)** | 17.3 | (1.7) | 16.9 | (1.8) | 16.9 | (1.7) | 17.3 | (2.4) |
| Skinfolds (mm) | | | | | | | | |
| Suprailiac | 20.1 | (10.4) | 21.4 | (8.0) | 19.0 | (7.8) | 18.2 | (9.7) |
| Mid-Thigh (A)** | 17.9* | (10.2) | 32.2 | (9.7) | 18.7 | (10.0) | 30.2 | (10.3) |
| Mid-Thigh (P)** | 18.1* | (11.1) | 30.8 | (10.4) | 17.6 | (11.4) | 29.3 | (10.4) |
| Abdomen | 24.6 | (11.6) | 26.3 | (9.7) | 26.1 | (10.7) | 28.8 | (13.9) |

*Note.* A = anterior; P = posterior; M = medial; L = lateral.

*$P < .05$, significant difference between sexes within generation sample.

**Average of lateral measurements from the left and right side of the body.

*** Measurements were taken only from the right side of the body.

Creative Health Products, Plymouth, MI.). Participants also underwent a full body DXA scan (GE Lunar Prodigy Advance: scan pixel resolution of 1.2 mm x 1.8 mm, mass resolution of 0.01 g/mm², scan time ~5 min.) while supine. Using enCORE™ software (2013, GE Healthcare, v. 15.00.362), the DXA scans were analyzed to determine participants' actual bilateral IC and GT soft tissue thicknesses (in centimeters) by manually measuring the horizontal distance between the most lateral aspect on the right and left IC and GT and the lateral edge of the soft tissue (i.e., the skin-air boundary [13, 17, 19]), respectively. A subset of 80 participants (i.e., 20 females and 20 males from each age group) were measured twice by the same investigator to quantify the intra-measurer reliability of the DXA scan measurements.

**Table 3. Description of anthropometric measurements taken and recorded to the nearest millimetre from the trunk, pelvis, and thigh segments.**

| Measurements | Segment | Description and landmarks |
|---|---|---|
| Lengths | Pelvis (A) | Vertical distance between the pubic symphysis and the most superior point of the iliac crests. |
| | Pelvis (L) | Vertical distance between the greater trochanter and the most superior point on the iliac crest. |
| | Thigh (L) | Distance between the superior iliac crest and the lateral aspect of the tibial plateau. |
| | Thigh (M) | Distance between the anterior level of the pubic symphysis and the medial aspect of the tibial plateau. |
| | Mid-Thigh | Distance around the thigh midway between the superior iliac crest and the tibial plateau. |
| Circumferences* | Hips | Horizontal distance around the hips at the level of the greater trochanters of the femurs. |
| | Waist | Horizontal distance around the trunk at the level midway between the last ribs and the most superior point of the iliac crests. |
| | Pelvis | Horizontal distance around the pelvis at the level of the superior aspects of the iliac crests. |
| | Upper Thigh | Distance around the thigh just inferior to the gluteal fold |
| | Mid-Thigh | Distance around the thigh. midway between the superior iliac crest and the tibial plateau. |
| Breadths/ Depths* | Waist (A-P) | Distance across the abdomen at the level of the waist circumference along the antero- posterior axis. |
| | Waist (M-L) | Distance across the abdomen at the level of the waist circumference along the medio-lateral axis. |
| | Pelvis (A-P) | Distance across the pelvis at the level of the sacral hiatus along the antero-posterior axis. |
| | Hip (M-L) | Distance across the hips at the level of the maximum circumference in the frontal plane. |
| | Upper Thigh (A-P) | Distance across the thigh just inferior to the gluteal fold. |
| | Max Thigh (M-L) | Distance across the thigh at the level of maximum circumference midway between the superior iliac crest and the tibial plateau. |
| | Max Thigh (A-P) | Distance across the thigh at the level of maximum circumference midway between the superior iliac crest and the tibial plateau. |
| Skinfolds** | Suprailiac | Oblique fold taken in line with the natural angle of the iliac crest immediately superior to the iliac crest. |
| | Mid Thigh (A) | Vertical fold on the anterior aspect of the thigh at the level of maximum circumference midway between the superior iliac crest and the tibial plateau. |
| | Mid Thigh (P) | Vertical fold on the posterior aspect of the thigh at the level of maximum circumference midway between the gluteal fold and the popliteal fossa with the subject lying prone. |
| | Abdomen | Vertical fold 2 cm to the right side of the umbilicus. |

*Note*. A = anterior; P = posterior; M = medial; L = lateral; A-P = antero-posterior; M-L = medio-lateral.

*All circumferences and breadths/depths measured after a normal exhalation. Participants were also instructed to stand with feet slightly narrower than shoulder width, in a normal, relaxed state.

**Skinfold locations parallel those from Jackson and Pollock (1978) [25].

## Data analysis and statistical procedures

The anthropometric measurements from participants and thickness measurements from the DXA scans were screened for invalid or missing values, as well as potential outliers (SD +3.29); if necessary, these values were replaced with the mean of their respective age and sex group

[26]. Differences between sexes were assessed via independent samples t-tests. To develop and test soft tissue thickness prediction equation accuracy for each age group, generation and validation samples were randomly created from the younger and older participant groups [21–24]: younger generation sample of 74 participants (37 males, 37 females) and validation sample of 26 participants (13 males, 13 females); older generation sample of 76 participants (38 males, 38 females) and validation sample of 25 participants (13 males, 12 females). Independent samples t-tests and Levene's test were used to assess the homogeneity of variance between the generation and validation samples for each age group. Normality of the data in the generation samples was evaluated using the ratio of skewness and kurtosis statistic to its corresponding standard error [27]; most of the variables were found to be normally distributed (i.e., did not exceed ± 1.96 at $P < 0.05$), with the exception of the lengths, skinfolds, and the upper thigh circumference measurement for the older population. To reduce the effect of multicollinearity, bivariate correlations were used to identify predictor variables that demonstrated high correlations ($r \geq 0.8$). This helped reduce the number of variables used in the regression models. All statistical tests were executed using SPSS (IBM SPSS Statistics, Version 23, IBM Corporation, Somers, NY). Intra-class correlation coefficients were computed to quantify intra-rater reliability of the DXA scan measurements.

Multiple linear stepwise regression was used to generate two sets of four prediction equations for the right and left IC and GT soft tissue thicknesses of the younger and older groups. On average, the soft tissue thicknesses measured from DXA for the right side of the body were approximately 3 mm larger than the left side across both the younger (IC: L = 2.13 ± 1.12 cm, R = 2.48 ±1.22 cm; GT: L = 3.53 ±1.46 cm, R = 3.84 ±1.49 cm) and older (IC: L = 3.71 ± 1.96 cm, R = 4.08 ±2.02 cm; GT: L = 4.44 ±1.97 cm, R = 4.71 ±2.04 cm) age groups, with the mean IC soft tissue thickness for the younger group being significantly different between the right and left sides. As a result, all bilateral variables were constrained to the right or left side of the body for their respective equations. Anthropometric data and participant characteristics (i.e., sex, height, body mass, and BMI) from the respective validation samples were then input into the generated equations to quantify the accuracy of the predicted tissue thicknesses [21–24]. Predicted soft tissue thicknesses and actual soft tissue thicknesses were compared using the absolute error, percent (%) error, and root-mean-squared error. The relationships between predicted and actual soft tissue thicknesses were illustrated using simple linear regression and scatterplots.

## Results

Data screening revealed two participants from the younger group (one male, one female) who were not representative of the population as several measurements exceeded the z-score cut-off of 3.29 (waist circumference, pelvis circumference, skin folds, etc.), and thus, were removed from the study. No significant differences were found, and variances were equal between the generation and validation samples for each age group for all mean participant anthropometric measurements and physical characteristics ($P > 0.05$). The reliability analysis revealed that the average measure intra-class correlation coefficients was > 0.997 ($P < 0.01$) for left and right IC and GT. Within the generation samples, significant differences in the mean anthropometric measurements were detected between sexes for 13 of 24 variables in the younger participants and 9 of 24 variables in the older participants ($P < 0.05$) (Tables 1 and 2), indicating that sex was a potential predictor for both populations. The number of predictor variables used to develop the IC and GT soft tissue thickness regression equations was reduced from the original 32 variables (Tables 1 and 2) to 10 (see footnote Table 4), because of the stepwise regression and bivariate correlation analyses conducted between the measurements.

**Table 4. Prediction equations for the left and right iliac crest (IC) and greater trochanter (GT) soft tissue thicknesses for the younger and older adult populations.**

| Anatomical Landmark and Side of Body | Eq. # | Adj. $R^2$ | SEE (cm) |
|---|---|---|---|
| **Younger** | | | |
| **Iliac Crest (IC)** | | | |
| $Y(right) = 1.871 + 0.046x_3 + 0.021x_4 + 0.106x_5 - 4.116x_2$ | 1 | 0.704 | 0.67 |
| $Y(left) = 3.921 + 0.017x_6 + 0.044x_3 - 4.474x_2 + 0.078x_7$ | 2 | 0.718 | 0.58 |
| **Greater Trochanter (GT)** | | | |
| $Y(right) = -6.176 + 0.038x_4 + 0.165x_5 - 1.522x_1$ | 3 | 0.749 | 0.77 |
| $Y(left) = -4.883 + 0.049x_6 - 1.392x_1 + 0.134x_7$ | 4 | 0.791 | 0.69 |
| **Older** | | | |
| **Iliac Crest (IC)** | | | |
| $Y(right) = -2.689 + 0.147x_8 - 1.288x_1 - 5.039x_2 + 0.031x_3$ | 5 | 0.825 | 0.82 |
| $Y(left) = -1.100 + 0.134x_8 - 1.313x_1 - 5.348x_2 + 0.030x_3$ | 6 | 0.852 | 0.72 |
| **Greater Trochanter (GT)** | | | |
| $Y(right) = -2.025 + 0.052x_4 + 0.241x_9 - 1.219x_1 + 0.082x_8 - 3.789x_2$ | 7 | 0.833 | 0.86 |
| $Y(left) = -1.644 + 0.047x_6 - 1.310x_1 + 0.074x_8 + 0.212x_{10} - 3.354x_2$ | 8 | 0.819 | 0.86 |

*Note*. $x_1$ = sex (f = 0, m = 1), $x_2$ = height (m), $x_3$ = abdomen skinfold (mm), $x_4$ = anterior mid thigh skinfold right (mm), $x_5$ = upper thigh circumference right (cm), $x_6$ = anterior mid thigh skin fold left (mm), $x_7$ = upper thigh circumference left (cm), $x_8$ = hip circumference, $x_9$ = max-thigh breadth right (M-L) (cm), $x_{10}$ = max-thigh breadth left (M-L) (cm)

SEE = Standard Error of the Estimate

Eight prediction equations were generated for estimating the bilateral soft tissue thicknesses overlying the IC and GT for the younger and older groups (Table 4). Equations for the younger group exhibited adjusted $R^2$ values ranging from 0.704 (right IC) to 0.791 (left GT), indicating slightly more explained variance for the GT than the IC soft tissue thicknesses. Corresponding adjusted $R^2$ values for the older group were more consistently grouped and ranged from 0.819 (left GT) to 0.852 (left IC). Across all equations, sex, height, abdomen skinfold, and hip circumference emerged as the main predictor variables, wherein sex, abdomen skinfold, and hip circumference were included in four equations and height in six equations. Standard errors in soft tissue thickness ranged from 0.58 cm to 0.77 cm and 0.78 cm to 0.86 cm for the younger and older groups, respectively. The root-mean-squared error values were highest at the IC for both the younger ($\leq$ 0.90 cm) and older ($\leq$ 0.87 cm) groups (Table 5). The prediction

**Table 5. Mean (± SD) predicted and actual (DXA) soft tissue thicknesses and errors from the younger and older validation sample (n = 26).**

| Anatomical Landmark | Actual (cm) | | Predicted (cm) | | Root-Mean-Squared Error (cm) |
|---|---|---|---|---|---|
| **Younger** | | | | | |
| Right IC | 2.47 | (1.20) | 2.72 | (0.82) | 0.86 |
| Left IC | 2.14 | (1.22) | 2.36 | (0.75) | 0.86 |
| Right GT | 4.02 | (1.32) | 3.85 | (1.32) | 0.49 |
| Left GT | 3.81 | (1.29) | 3.53 | (1.09) | 0.71 |
| **Older** | | | | | |
| Right IC | 3.95 | (2.20) | 3.84 | (2.16) | 0.86 |
| Left IC | 3.47 | (2.26) | 3.53 | (2.04) | 0.87 |
| Right GT | 4.75 | (1.87) | 4.63 | (2.05) | 0.81 |
| Left GT | 4.48 | (1.85) | 4.29 | (1.87) | 0.75 |

*Note*. IC = iliac crest; GT = greater trochanter.

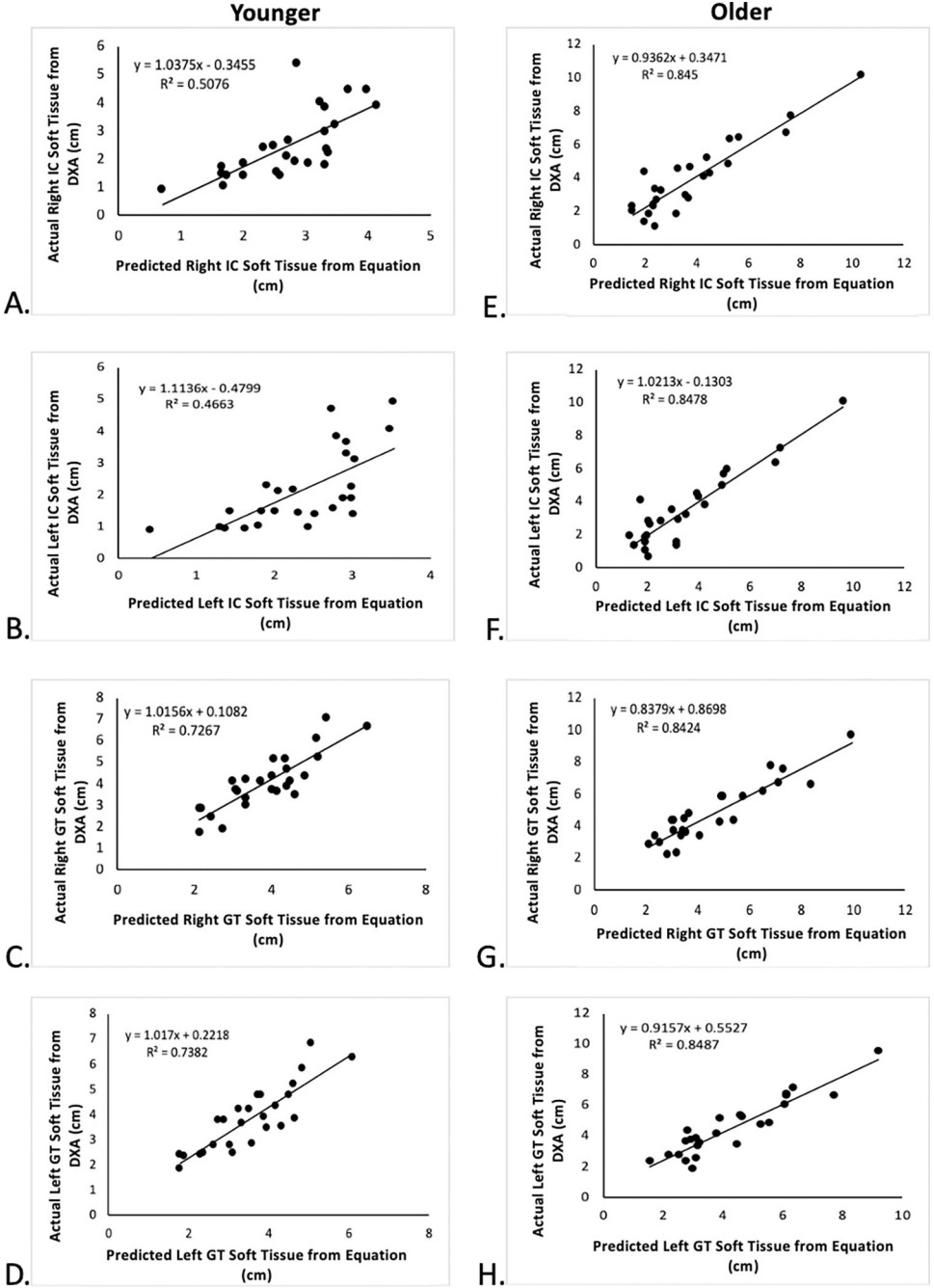

**Fig 1. Relationship between predicted and actual soft tissue thicknesses.** Soft tissue thickness of the younger population's right IC (A), left IC (B) right GT (C), left GT (D); and the older population's right IC (E), left IC (F) right GT (G), left GT (H). DXA = Dual X-Ray Absorptiometry; cm = centimeters; IC = Iliac Crest; GT = Greater Trochanter.

equations created for the older population had higher $R^2$ values (0.819 to 0.852) in comparison to the younger population (0.704 to 0.791). Pearson correlations between the predicted and actual soft tissue thicknesses measured by DXA were statistically significant for all equations and high in general (Fig 1), with $R^2$ values ranging from 0.466 to 0.738 for the younger group and 0.842 to 0.848 for the older group.

## Discussion

This study presents regression equations using anthropometric measurements to predict soft tissue thicknesses overlying the left and right IC and GT of healthy younger and older adults. In general, these equations exhibited strong adjusted $R^2$ values (($\geq 0.704$—younger) and ($\geq 0.819$—older)). Correlations between the predicted and actual soft tissue thicknesses were all statistically significant ($P < 0.01$), with errors ranging from 0.58 to 0.75 cm and 0.65 to 0.69 cm for the younger and older groups, respectively. This research expands on previous work [17, 19–24] and provides researchers and clinicians with an accurate, cost-effective, and relatively quick method for quantifying soft tissue thicknesses overlying the hip without the restrictions imposed by DXA scanning.

Overall, the older group equations resulted in higher adjusted $R^2$ values compared to the younger group. A potential explanation for this could be that the data were more normally distributed for the older group. A majority of the younger participants were recruited from a population of relatively fit kinesiology students at the affiliated university. This may have contributed to the minor skewing of the data in this group. It was encouraging that the prediction equations for the older adults were more accurate, given that they are at greater risk for hip fractures generally [28, 29].

The GT prediction equations accounted for more variance in soft tissue thickness than the IC equations. The difficulty associated with landmarking the most lateral point of the IC, combined with pixelation of the DXA scans, may have contributed to this result. However, the soft tissue thickness specifically overlying the GT has been documented as an important factor for hip fractures [3, 13, 14], while knowing the soft tissue thickness at the IC has the potential to reduce sport-related injuries such as hip pointers [6]. Consequently, it is suggested that the greater predictive capabilities of the GT soft tissue thickness equations are likely more relevant to hip fracture risk predictions than those for IC [3, 13, 14].

Skinfold measurements proved to be important predictor variables for generating the soft tissue thickness equations in this study, with at least one skinfold measurement present in each of the eight equations. Compared to other anthropometric measurements (e.g., waist circumference, sagittal abdominal diameter, etc.), the intra-rater reliability associated with skinfold measurements has been found to be lower [30]. Despite this potential limitation, abdominal skinfold measurements (an important predictor in all IC equations) have been shown to be the least variable compared to other skinfold measurements [30]. Regardless, when using the developed equations in practice, measurers should be well trained to reduce the variability of skinfold measures as much as possible.

There was a wide range in participant ages in this study (16 to 65 years), but the group was relatively young compared to some previous work. For example, Choi et al. (2015) found that the differences in hip soft tissue thicknesses between a younger group (19–30 years) and an older group (65–81 years) were not statistically significant. Therefore, future research should evaluate the predictive capability of the equations generated for the older group in the current study for participants older than 65 years. Although age-related differences in soft tissue stiffness and damping could increase the inherent risk of hip fractures [13, 28], more recent research has shown that trochanteric soft tissue thickness is a more critical factor than soft tissue stiffness in terms of hip fracture risk [16].

BMI was the only predictor variable in the regression equations used by Dufour et al. (2012). However, BMI was not used in the current study because it was highly correlated with other variables that proved to be more important predictors of trochanteric soft tissue thickness. Bivariate correlation analyses were performed to highlight which intended predictor variables were highly correlated, with the intent of reducing the effect of multicollinearity. In the

younger group, BMI and height were too highly correlated with each other to be included together. In the older group, BMI was too highly correlated with hip circumference. However, height and hip circumference in the younger and older groups, respectively, were more highly correlated with the soft tissue thicknesses compared to BMI, so they remained in the multivariate analysis.

The predictive equations presented herein do not require measurements derived from DXA scans, thereby reducing the cost associated with using the equations and eliminating the radiation exposure to participants. In comparison, to utilize the trochanteric soft tissue prediction equation presented previously by Schacter and Leslie (2014), two measurements from a regional DXA scan of the spine and hip are required (i.e., spine and hip average thicknesses). In addition, the current prediction equations account for more variance in trochanteric soft tissue thickness [$R^2$ values ranging from 0.704 to 0.852 (standard error of estimate < 0.86 cm), compared to an $R^2$ value of 0.6 (standard error of estimate = 1.35 cm) reported by Schacter and Leslie (2014)], and having separate equations for the IC and GT soft tissue thicknesses for both the right and left sides of the body, increases the specificity of the predictions that can be made. This offers clinicians and researchers who study the role that hip soft tissues play in reducing the risk of hip fracture from falling with an encouraging alternative to scanning technologies.

The findings of the present study suggest that soft tissue thicknesses overlying the IC and GT can be accurately estimated using surface anthropometric measurements from healthy adults across a range of ages. Compared to previous methods used to predict soft tissue thicknesses at the hip (e.g., hip circumference, percent body fat, spinal measurements, BMI), the current prediction equations are more practical to use and more accurate on average, than those previously presented. Clinicians can use the equations to identify individuals at greater risk of hip fractures who may require preventative measures such as hip protectors and fall prevention programs. Future work on individuals over the age of 65 years should be considered to expand the generalizability of this useful tool.

## Supporting information

**S1 File. Statistical output for the younger group.**
(PDF)

**S2 File. Statistical output for the older group.**
(PDF)

## Acknowledgments

Thank you to the Diagnostic Imaging Department at Windsor Regional Hospital for the use of their facility, equipment, and technical support.

## Author Contributions

**Conceptualization:** Andrew C. Laing, David M. Andrews.

**Data curation:** David M. Andrews.

**Formal analysis:** Claudia M. Town, Danielle L. Gyemi, Zoe Ellis.

**Funding acquisition:** David M. Andrews.

**Investigation:** Danielle L. Gyemi, Charles Kahelin.

**Methodology:** Danielle L. Gyemi, Charles Kahelin.

**Project administration:** David M. Andrews.

**Resources:** David M. Andrews.

**Supervision:** Danielle L. Gyemi, David M. Andrews.

**Visualization:** Claudia M. Town, Danielle L. Gyemi, David M. Andrews.

**Writing – original draft:** Claudia M. Town.

**Writing – review & editing:** Claudia M. Town, Danielle L. Gyemi, Zoe Ellis, Charles Kahelin, Andrew C. Laing, David M. Andrews.

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
