## [Decision Letter · Decision Letter 0]

14 Feb 2023

PONE-D-23-00184Predicting soft tissue thicknesses overlying the iliac crests and greater trochanters of younger and older adultsPLOS ONE

Dear Dr. Town,

Thank you for submitting your manuscript to PLOS ONE. After careful consideration, we feel that it has merit but does not fully meet PLOS ONE’s publication criteria as it currently stands. Therefore, we invite you to submit a revised version of the manuscript that addresses the points raised during the review process.

We look forward to receiving your revised manuscript.

Kind regards,

Mohamed El-Sayed Abdel-Wanis, Ph.D.

Academic Editor

PLOS ONE

Journal Requirements:

**Additional Editor Comments:**

Marvelous work indeed. Thank you for rigorous statistical analysis and efforts to extract these data to build up a solid evidence.

Abstract

Adequate.

The introduction section is lengthy. It needs to be shortened

The discussion is robust and well-cited.

Reviewers' comments:

Reviewer's Responses to Questions

**Comments to the Author**

1. Is the manuscript technically sound, and do the data support the conclusions?

Reviewer #1: Yes

2. Has the statistical analysis been performed appropriately and rigorously? 

Reviewer #1: Yes

3. Have the authors made all data underlying the findings in their manuscript fully available?

Reviewer #1: Yes

4. Is the manuscript presented in an intelligible fashion and written in standard English?

Reviewer #1: Yes

5. Review Comments to the Author

Reviewer #1: I congratulate the authors for this well-done and well-presented study.

I only have one comment: in Tables 1 and 2, the mean values of the skin fold should be presented in mm, not in cm as mentioned in the two tables.

Good luck

6. PLOS authors have the option to publish the peer review history of their article (what does this mean?). If published, this will include your full peer review and any attached files.

Reviewer #1: **Yes: **Tarek A. Abulezz

---

## [Author Response · Author response to Decision Letter 0]

17 Feb 2023

Editor: Please ensure that your manuscript meets PLOS ONE's style requirements, including those for file naming. Please provide additional details regarding participant consent. In the ethics statement in the Methods and online submission information, please ensure that you have specified what type you obtained (for instance, written or verbal, and if verbal, how it was documented and witnessed). If your study included minors, state whether you obtained consent from parents or guardians. If the need for consent was waived by the ethics committee, please include this information. Please include captions for your Supporting Information files at the end of your manuscript, and update any in-text citations to match accordingly. Please review your reference list to ensure that it is complete and correct. If you have cited papers that have been retracted, please include the rationale for doing so in the manuscript text, or remove these references and replace them with relevant current references. Any changes to the reference list should be mentioned in the rebuttal letter that accompanies your revised manuscript. If you need to cite a retracted article, indicate the article’s retracted status in the References list and also include a citation and full reference for the retraction notice. The introduction section is lengthy. It needs to be shortened. 

Response: Thank you for your comments. We have included all the suggestions in our revisions. 

Reviewer #1 

I only have one comment: in Tables 1 and 2, the mean values of the skin fold should be presented in mm, not in cm as mentioned in the two tables.

Response: Thank you for your comment. We have revised the tables to present skin fold measurements in mm.

---

## [Editor Report · Decision Letter 1]

1 Mar 2023

Predicting soft tissue thicknesses overlying the iliac crests and greater trochanters of younger and older adults

PONE-D-23-00184R1

Dear Dr. Town 

We’re pleased to inform you that your manuscript has been judged scientifically suitable for publication and will be formally accepted for publication once it meets all outstanding technical requirements.

Kind regards,

Mohamed El-Sayed Abdel-Wanis, Ph.D.

Academic Editor

PLOS ONE

---

## [Editor Report · Acceptance letter]

6 Mar 2023

PONE-D-23-00184R1 

Predicting soft tissue thicknesses overlying the iliac crests and greater trochanters of younger and older adults 

Dear Dr. Town:

I'm pleased to inform you that your manuscript has been deemed suitable for publication in PLOS ONE. Congratulations! Your manuscript is now with our production department. 

Kind regards, 

on behalf of

Prof. Dr Mohamed El-Sayed Abdel-Wanis 

Academic Editor

PLOS ONE